# Two-Channel OTDM System for Data-Center Interconnects: A Review

**DOI:** 10.3390/s23135908

**Published:** 2023-06-26

**Authors:** Sunghyun Bae, Hyeon-June Kim

**Affiliations:** 1Department of Electronics Engineering, Kangwon National University, Samcheok 25913, Republic of Korea; baesh@kangwon.ac.kr; 2Department of Semiconductor Engineering, Seoul National University of Science and Technology, Seoul 01811, Republic of Korea

**Keywords:** data-center interconnect, optical time-division multiplexing, pulse-amplitude modulation

## Abstract

It has been proposed to implement the >100 Gb/s data-center interconnects using a two-channel optical time-division multiplexed system with multilevel pulse-amplitude modulation. Unlike the conventional four-channel optical time-division multiplexed system which requires an expensive narrow pulse, the two-channel system can be implemented cost-effectively using a wide pulse (which can be simply generated using a single modulator). The two-channel system is expected to be practically available using an integrated transmitter in a chip due to the recent advances in photonics-integrated circuits. This paper reviews the current stage of research on a two-channel optical time-division multiplexed system and discusses possible research directions. Furthermore, it has been demonstrated that 200 Gb/s signals can be generated by using modulators with only 17.2 GHz bandwidth. Therefore, the use of the phase-alternating pulse can make the multiplexed signal robust to chromatic dispersion, enabling the 200 Gb/s 4-level pulse-amplitude-modulated signal to be transmitted over 1.9 km of standard single-mode fiber.

## 1. Introduction

Data centers have become essential infrastructure for society due to the rapid exploration of data traffic through high-definition video and audio streaming services, cloud services, and social networks [1,2,3,4]. The 400 gigabit Ethernet will employ multiple 50 or 100 Gb/s/λ 4-level pulse-amplitude modulation (PAM-4) signals as specified in the Ethernet standard [5,6]. Considering the steep growth in traffic [7,8], it is necessary to develop a way to implement Ethernet beyond 400 gigabits (such as 800 gigabit or 1.6 terabit Ethernet) implemented by multiple ≥100 Gb/s/λ signals soon [3,9,10]. Since the use of a cost-effective modulation scheme is still preferred [1,3,11], various studies have applied PAM-8 to implement beyond 400 gigabit Ethernet, which can be made cost-effective by using a 3-bit digital-to-analog converter (DAC) and reducing the bandwidth requirements of components [2,12,13]. However, PAM-8 is sensitive to impairments caused by the limited bandwidth of transceivers and chromatic dispersion (CD) [14]. Thus, PAM-8 may require various techniques, such as Tomlinson–Harashima precoding [15], a machine-learning-based equalizer [16], or maximum-likelihood sequence estimation [13,14], to improve the signal-to-noise and distortion ratio (SINAD), which can be considered extremely complex and cost-intensive for data-center interconnects (DCIs). Consequently, the use of the probabilistically shaped PAM-8 signal has also been considered for stable operation by reducing the spectral efficiency to less than three [16,17,18]. Therefore, it is unlikely that the modulation level of PAM exceeds eight, which makes the implementation of high-speed DCIs based on intensity-modulation and direct-detection (IM/DD) systems challenging.

J. Verbist et al. proposed using optical time-division multiplexing (OTDM) to increase the operating speed of DCIs [19]. They demonstrated the transmission of a 208 Gb/s OTDM-PAM-4 signal using a four-channel OTDM system [19], which was actively researched in the 1990s and 2000s [20]. Although such a system is more complex than the conventional IM/DD system that uses a single pair of transceivers, it makes high-speed operation possible by overcoming the insufficient bandwidth of the modulator [21]. An optical pulse source with a high repetition rate is required on the transmitter side to implement the four-channel OTDM system [20]. The optical pulse should have a high extinction ratio (ER) and a narrow pulse width, commonly generated using an actively mode-locked laser consisting of a Fabry–Perot resonator, a gain medium, and a modulator [20,22]. Among the multiple longitudinal modes determined by the cavity, only the harmonics of a certain frequency (determined by the modulation frequency) can be locked. However, stabilizing the pulse with a high repetition rate is challenging because it requires high-precision timing distribution systems [23,24]. An alternative method involves the use of a comb generator, which typically consists of an intensity modulator, a phase modulator, and a dispersion compensating fiber (DCF) [25,26]. However, this method requires high power consumption for comb generation and a few hundred meters of DCF to increase the ER and reduce the pulse width [26].

Recently, we proposed the implementation of high-speed DCIs with a two-channel OTDM-PAM-*N* signal [21]. In this system, the pulse source can be simply generated using a modulator and a sinusoidal electrical clock. In the previously proposed two-channel OTDM system, the OTDM signal was generated using the sinusoidal optical pulse and detected using a single photodiode (PD), which is valid because a PD has a larger bandwidth than a modulator [27,28]. The crosstalk between two modulated signals can be compensated using a 2×2 multiple-input multiple-output (MIMO) equalizer realized through matured digital signal processing (DSP), enabling the use of sinusoidal pulses with low ER.

This paper reviews recent research on two-channel OTDM systems. The remainder of the paper is organized as follows: Section 2 reviews the principles of a two-channel OTDM system implemented using a constant-phase pulse source. Section 3 reviews an OTDM system implemented using a phase-alternating pulse source to increase the transmission distance. Section 4 discusses the possible research direction for the commercialization of a two-channel OTDM systems realized in short-reach interconnects. Section 5 presents the conclusions.

## 2. Two-Channel OTDM System with Constant-Phase Pulse

### 2.1. System Architecture

An optical pulse with a repetition rate of *B* is necessary to generate a two-channel OTDM signal with a baud rate of 2B; the pulse width need not be narrow. A sinusoidally modulated optical pulse with a repetition rate of *B* can be generated using a Mach–Zehnder modulator (MZM) (biased at the quadrature point) with an electrical clock at a frequency of *B*, as illustrated in Figure 1. The pulse is divided into two parts using an optical coupler, with each part modulated by an electrical PAM signal to generate a return-to-zero (RZ) PAM signal. One signal is delayed by 1/(2B) to realize time-multiplexing in the optical domain, and the other signal is passed through a domain shifter (DS) with a quarter-wave plate or a phase shifter to avoid beat components during multiplexing. The generated OTDM signal with a baud rate of 2B is amplified by a semiconductor optical amplifier (SOA), detected using a single PD, demultiplexed in the electrical domain, and equalized using a 2×2 MIMO equalizer.

### 2.2. OTDM System Using the Symbol-Spaced MIMO Equalizer

#### 2.2.1. Transmission Experiments

Figure 2 shows the experimental setup used to transmit a 150 Gb/s OTDM signal generated using the sinusoidal optical pulse [21]. For pulse generation, the output of a 1550 nm laser was modulated using a commercial LiNbO_3_ MZM1 (3-dB bandwidth = 17.2 GHz). The MZM was biased at the quadrature point and driven by a 25 GHz electrical clock. The pulse with a 7.7-dB ER was modulated using an MZM2 to generate the 75 Gb/s RZ-PAM-8 signal. The electrical PAM-8 signal was generated by a 3-bit DAC and three outputs from a pulse-pattern generator (PPG). The MZM2 was driven through a push–pull operation using 75 Gb/s differential pair signals from the DAC. Instead of SOAs, erbium-doped fiber amplifiers (EDFAs) were used to compensate for the insertion losses of MZMs. The RZ signal was split into two parts using a polarization beam splitter (PBS) and combined again using a polarization beam combiner (PBC). A proper delay line was inserted in one of the paths to realize decorrelation and accurate timing alignment (for multiplexing). The generated signal was transmitted over a 1.9 km standard single-mode fiber (SSMF) and detected by a receiver consisting of a PIN PD (3 dB bandwidth = 70 GHz) and an electrical amplifier (3-dB bandwidth = 50 GHz). After the signal was captured using a real-time oscilloscope (RTO; sampling rate = 160 GSample/s), it was demultiplexed and equalized using a 2×2 MIMO equalizer consisting of 7-tap symbol-spaced feed-forward equalizers (FFEs).

Figure 3a shows the bit-error rate (BER) curves of the 150 Gb/s OTDM signal depending on the received optical power [21]. The receiver sensitivities (measured at a BER of $3.8\times 10^{-3}$ determined by a forward error correction (FEC)) of the two demultiplexed signals were −5.5 dBm and −5.1 dBm, respectively. A penalty of approximately 2.0~2.9 dB was observed compared with the 75 Gb/s PAM-8 signal generated by a single MZM and equalized by a 6-tap symbol-spaced FFE on the receiver side due to the doubled bit rate and the decreased signal-to-noise ratio (SNR) owing to the finite ER of the pulse. Figure 3b shows the BERs of the OTDM-PAM-8 signal depending on the bit rate. The number of taps for each FFE constituting the MIMO equalizer was increased to 11. The resulting signals exhibited speeds up to 174 Gb/s and could reach an FEC limit of BER = $3.8\times 10^{-3}$. Notably, the bandwidth of the MZMs was only 17.2 GHz.

#### 2.2.2. Roles of the MIMO Equalizer

A numerical simulation was performed to evaluate the role of the MIMO equalizer [21]. The following assumptions were made to ensure consistency with the experiment: (1) the bit rate of the OTDM signal was 150 Gb/s; (2) the sinusoidal pulse was generated using a laser diode (LD) and an MZM; (3) the two divided optical pulses were converted into the RZ-PAM-*N* (N=2,4,8) signals by the two MZMs (biased at quadrature points); (4) the peak-to-peak amplitude (ΔV) of the electrical PAM signals were $0.6V_{\pi }$ ($V_{\pi }$: half-wave voltage); (5) all the MZMs had sufficient bandwidth (i.e., 3 dB bandwidth = 75/N GHz); (6) the OTDM signal was detected using a receiver comprising a PIN PD (3 dB bandwidth = 150/N GHz, responsivity = 1 A/W), 50Ω load resistor at $20{\;}^{\circ }C$, and an electrical amplifier (noise figure = 6 dB); and (7) the FEC limit was set to a BER of $3.8\times 10^{-3}$.

The MIMO equalizer can successfully eliminate crosstalk between RZ signals caused by the finite ER of the optical pulse. In this simulation, the MIMO equalizer consisted of symbol-spaced 11-tap FFEs. When the MIMO equalizer was not used, a 2 dB power penalty was observed when the pulse ER was 6 dB, 9 dB, and 14 dB, respectively, compared to the case when the pulse ER was 25 dB (as shown in Figure 4). Since the penalty was primarily caused by crosstalk, the higher-order modulated signal was more sensitive. However, when the MIMO equalizer was used, a 2 dB penalty was observed when the pulse ER was 5 dB, 5 dB, and 6 dB, respectively. The receiver sensitivity was less sensitive to the modulation format because the crosstalk was effectively compensated. The residual penalty was primarily caused by the reduced ER of the OTDM signal and can be expressed as follows: (1)$$\mathrm{Penalty}\left(\mathrm{dB}\right)=10\log\left(\frac{\epsilon_{\mathrm{pulse}}+1}{\epsilon_{\mathrm{pulse}}}\right),$$
where $\epsilon_{\mathrm{pulse}}$ denotes the pulse’s ER. The results indicated that the two-channel OTDM system was operable as long as the ER of the pulse was >5 dB [21].

The MIMO equalizer effectively operated even when there was imperfect multiplexing. During the simulation, the pulse ER was set to 10 dB. A power mismatch can be caused by either the mismatched loss of waveguides propagating two RZ signals in the transmitter or the polarization-dependent loss (when two RZ signals in different polarization domains are combined). Figure 5 displays the co-plots (dashed lines) of receiver sensitivities as a function of the power difference between two multiplexed signals. The residual penalty can be interpreted as a crosstalk penalty due to the imperfections of the MIMO equalizer. Although the RZ1 signal with a lower power was more sensitive to power mismatch, the residual penalty was as low as <0.2 dB (which is negligible) when the power mismatch was <1.5 dB. Moreover, a timing mismatch could be caused by either the imperfect delay line in the transmitter or the polarization-mode dispersion (when the two RZ signals in different polarization domains were combined). The timing mismatch can be expressed as follows: (2)$$\mathrm{Timing}\;\mathrm{mismatch}=\frac{|\tau -T_{s}|}{T_{s}},$$
where τ denotes the delay time caused by the optical delay line, and $T_{s}$ denotes the symbol period of the OTDM signal. When the timing mismatch was >16%, a 1 dB penalty was observed, as shown in Figure 5b. The MIMO equalizer can be implemented with half-symbol-spaced FFEs to eliminate the penalty.

### 2.3. OTDM System Using a Multilayer Equalizer

A multilayer equalizer for the two-channel OTDM system was proposed to reduce the BER floor (crucial when a higher-order modulation-formatted signal is used), as shown in Figure 6a [29]. The optical transmitter is identical to that shown in Figure 1. However, the demultiplexer is implemented with half-symbol-spaced equalizers. Thus, this system uses a multilayer equalizer consisting of a demultiplexer (as the first layer) and a MIMO equalizer (as the second layer). The structure of the demultiplexer consists of two half-symbol-spaced FFEs, as illustrated in Figure 6b. The input–output relationship can be expressed as follows: (3)$$\begin{array}{cc} x_{\mathrm{out},1}\left[k\right] & =h_{1}^{\left(k\right)}x_{\mathrm{in}}{\left[2k-1\right]}^{T}\\ x_{\mathrm{out},2}\left[k\right] & =h_{2}^{\left(k\right)}x_{\mathrm{in}}{\left[2k\right]}^{T}, \end{array}$$
where $x_{\mathrm{in}}\left[k\right]$ denotes a sequence consisting of half-symbol-spaced samples (i.e., $x_{\mathrm{in}}\left[k\right]=$ [$x_{\mathrm{in}}\left(kT_{s}+D_{1}\right)$, $x_{\mathrm{in}}\left(\left(k+0.5\right)T_{s}+D_{1}\right)$, $x_{\mathrm{in}}\left(\left(k+1\right)T_{s}+D_{1}\right)$, …, $x_{\mathrm{in}}\left(\left(k+L-0.5\right)T_{s}+D_{1}\right)$], $D_{1}$: An arbitrary constant representing a delay), and the superscript ‘*T*’ represents the transpose. $h_{1}^{\left(k\right)}$ and $h_{2}^{\left(k\right)}$ are alternatively applied to the OTDM signal to reconstruct the RZ1 signal, $x_{\mathrm{out},1}\left[k\right]$, and RZ2 signal, $x_{\mathrm{out},2}\left[k\right]$, respectively [29]. $h_{1}$ and $h_{2}$ can be interpreted as matched filters that maximize the SINAD. Furthermore, errors due to power and timing mismatches can be compensated for by separating $h_{1}$ and $h_{2}$.

While estimating the optimal sampling point using the FFE-based demultiplexer, the BER can be further reduced using an improved MIMO equalizer. The MIMO equalizer may consist of second-order Volterra equalizers, where the input–output relationship can be expressed as follows: (4)$$\begin{array}{cc} y[k] & =w_{0}+\sum_{k_{1}=-D_{2}}^{-D_{2}+M_{1}-1}w_{1}\left(k_{1}\right)x\left[k+k_{1}\right]+\\ \; & +\sum_{k_{2}=-D_{2}}^{-D_{2}+M_{1}-1}\sum_{k_{3}=k_{2}}^{\min(-D_{2}+M_{1}-1,k_{2}+M_{2}-1)}w_{2}\left(k_{2},k_{3}\right)x\left[k+k_{2}\right]x\left[k+k_{3}\right] \end{array}$$
where $D_{2}$ denotes a constant representing a delay, $M_{1}$ and $M_{2}$ denote constants determining the tap length, $w_{r}$ denotes the *r*-th order filter coefficient, and x[k] and y[k] represent the *k*-th input and output of the equalizer, respectively [29]. A parameter vector $M=[M_{1},M_{2}]$ can be introduced for convenience.

#### Experimental Results Obtained Using a Multilayer Equalizer

Using the data received from the setup shown in Figure 2, it was verified that the multilayer equalizer could improve BER performance. A time demultiplexer made up of two 6-tap half-symbol-spaced FFEs and a simplified symbol-spaced nonlinear MIMO equalizer made up the receiver. The 3 dB bandwidth of the receiver was set to 25 GHz. Figure 7a shows the BERs depending on the sampling position when using a symbol-spaced demultiplexer and a MIMO equalizer [29]. The received optical power was −1 dBm under the back-to-back condition. When the symbol-spaced demultiplexer was used, the $-T_{s}/4$ offset in the sampling position caused a 3.4-fold increase in BER (from $5.8\times 10^{-4}$ to $2.0\times 10^{-3}$). Notably, determining the optimal sampling position was challenging when using a higher-order modulation-formatted signal and a pulse with a finite ER, as shown in Figure 7c. In addition, the nonlinear MIMO equalizer with M=[12,12] could not improve the BER performance, likely because of the unsuitable time delay of the OTDM transmitter. By contrast, when using the multilayer equalizer, the BER performance was slightly affected by the sampling position. Subsequently, the nonlinear MIMO equalizer could effectively lower the BER, as shown in Figure 7b.

Figure 8a shows the BER curves of the 150 Gb/s OTDM-PAM-8 signals under the back-to-back condition when using a symbol-spaced demultiplexer with a MIMO equalizer [29]. The nonlinear MIMO equalizer could not improve the BER performance, consistent with the results shown in Figure 7a. Instead, the performance deteriorated because of overfitting. However, when the multilayer equalizer was used, the nonlinear MIMO equalizer with either M=[12,12] or [12,0] could improve the BER more effectively than the linear equalizer, as shown in Figure 7b, likely because it compensated for the impairments due to the unsuitable time delay of the transmitter. Therefore, a BER lower than $10^{-3}$ could be achieved using the multilayer equalizer (consisting of an FFE-based demultiplexer and a nonlinear MIMO equalizer) even after transmission over a 1.9 km SSMF.

## 3. Two-Channel OTDM System with Phase-Alternating Pulse

### 3.1. Phase-Alternating Pulse Generation

Figure 9a shows the structure of the OTDM transmitter that uses a phase-alternating pulse generated by an MZM. The MZM1 is biased at the null point and modulated using an electrical clock with a frequency of 0.5B and ΔV of $2V_{\pi }$. Then, a phase-alternating pulse with a 66% duty ratio is generated, as shown in Figure 9b. When generating the OTDM signal using the phase-alternating pulse, the maximum power of the transition region is larger than that of the symbol center due to the pulse’s large duty ratio, as shown in Figure 9c. Since the optical field of the phase-alternating pulse has a zero mean, the carrier of the phase-alternating pulse disappeared—resulting in a carrier-suppressed RZ (CSRZ) pulse. The carrier of the OTDM signal generated by the CSRZ pulse is also suppressed, as shown in Figure 9d. While the phase-alternating pulse has a broader pulse width compared with the constant-phase pulse, its optical spectrum is narrower, as shown in Figure 9d,g, making the generated OTDM signal robust to CD. Furthermore, the CSRZ pulse is generated using the half-rated electrical clock. Consequently, the required bandwidth for generating the optical pulse decreases.

Alternatively, a CSRZ pulse can be generated using a phase modulator (PM) and a periodic optical filter [30]. When the light is phase-modulated using an electrical clock with a frequency of 0.5B, multiple harmonics are generated because of the nonlinearity of the PM. By applying an optical filter with a free-spectral range of *B* to suppress the carrier frequency (as shown in Figure 10b), a CSRZ pulse with a repetition rate of *B* is generated, as shown in Figure 10c. The resulting CSRZ pulse and a half-rated electrical clock are used to generate an OTDM signal with a baud rate of 2B. The optimal ΔV of the PM is $1.2V_{\pi }$. In this method, the pulse can be generated without requiring an auto-bias controller.

As illustrated in Figure 9d,g, the phase-alternating pulse makes the OTDM signal robust to CD. Numerical simulations were performed to identify the robustness to CD from eye diagrams of 200 Gb/s OTDM-PAM-4 signals. A constant-phase and a phase-alternating pulse were generated using an MZM, as illustrated in Figure 1 and Figure 9a, respectively. The following assumptions were made: (1) the light had a wavelength of 1550 nm; (2) the pulse ER was infinite; (3) the two generated (CS)RZ-PAM-4 signals in different polarization domains were combined; (4) the CD parameter of the SSMF was 16 ps/(nm·km); and (5) the PMD parameter was 0.5 ps/${\mathrm{km}}^{0.5}$. Each level of the OTDM signal generated using a constant-phase pulse was not distinguishable after 1 km transmission, but each level generated using a phase-alternating pulse was distinguishable after 1 km transmission, as shown in Figure 11. The phase-alternating pulse made the OTDM signal robust to CD because the adjacent bits in each multiplexed channel were subjected to destructive interference.

### 3.2. Experimental Results

The transmission performance of the 200 Gb/s OTDM signal generated using the CSRZ pulse (i.e., phase-alternating pulse) was compared with that generated using the RZ pulse (i.e., constant-phase pulse). The 1549.9 nm light was phase-modulated with a 25 GHz electrical clock (ΔV = 0.55$V_{\pi }$) and sent to a 25/50 GHz optical interleaver (as a periodic filter) to generate the CSRZ pulse, as sown in Figure 12. When generating the RZ pulse, the ΔV of the electrical clock was increased to 0.75$V_{\pi }$, and another port of the interleaver was used. Then, the (CS)RZ PAM-4 signal was generated by modulating an MZM with a 100 Gb/s electrical PAM-4 signal (generated using a 92-GSample/s arbitrary waveform generator (AWG)). The (CS)RZ PAM-4 signal was split into two parts, which combined again after a proper time delay to generate the 200 Gb/s OTDM signal. The two (CS)RZ signals were in different polarization domains. The OTDM signal was amplified using an EDFA, transmitted over an SSMF, and detected using a 70 GHz PD. After the data were captured using an RTO at 160-GSample/s, a symbol-spaced 2×2 MIMO equalizer was implemented with symbol-spaced 30-tap FFEs.

Figure 13 shows the measured BER curves of the 200 Gb/s OTDM-PAM-4 signal. As illustrated in Figure 11b, the OTDM signal generated using the constant-phase RZ pulse was significantly deteriorated by the CD arising from the 1 km SSMF (in which the power penalty was measured to be as large as 2.6 dB). On the other hand, the OTDM signal generated using the phase-alternating CSRZ pulse was marginally affected by the CD arising from the 1 km SSMF, as shown in Figure 11d. In addition, the signal could be transmitted over a 1.9 km SSMF with the FEC limit set to BER $=3.8\times 10^{-3}$.

## 4. Discussion

It is expected that DCIs will be implemented with the 100, 200, and 400 Gb/s/λ IM/DD systems. Considering the limited bandwidth of commercially available transmitters, it is challenging to achieve high-speed DCIs operating at ≥100 Gb/s/λ with a single modulator. The use of the two-channel OTDM system enables sidestepping the insufficient bandwidth problem. It was demonstrated that the 200 Gb/s/λ PAM-4 signal can be implemented with 17.2 GHz MZMs. We believe that the 400 Gb/s/λ can also be realized using commercially available 40 GHz MZMs.

An integrated transmitter on a chip should be developed to practically deploy a two-channel OTDM system for DCIs. It was previously demonstrated that this system could be implemented using a transmitter integrated into a silicon photonics (SiP) chip, as shown in Figure 14 [21]. The 64 Gb/s OTDM-PAM-4 signal could be generated using a SiP chip consisting of two optical couplers, two MZMs, an optical delay line, and a thermal phase shifter (TPS) (as a DS) [21]. More recently, the 300 Gb/s OTDM PAM-8 signal was generated using a SiP transmitter and successfully transmitted over the 500 m SSMF [31]. However, the previously developed SiP transmitter [21,31] did not contain an optical pulse source due to its high insertion loss. In the future, it will be necessary to implement the OTDM system with the use of a SiP transmitter containing an optical pulse source. Furthermore, it is also necessary to develop a SiP OTDM transmitter integrating a polarization rotator (used as a DS) and a PBS [32]. Indium phosphide (InP) [28] and lithium niobate [33] -based modulators are also good candidates for the integrated photonics chip.

The optical pulse source with a high repetition rate is a key component of the two-channel OTDM system. A technique wherein an MZM is biased at a sub-quadrature point was proposed to increase the ER of the optical pulse with a high repetition rate [34]. This technique could generate a 50 GHz optical pulse with 14.9 dB ER even when using a 17.2 GHz MZM, enabling the transmission of the 300-Gb/s OTDM-PAM-8 signal and a soft-decision FEC with 20% overhead (the FEC limit was BER $=2.7\times 10^{-2}$). The generated signal could be transmitted over a 1 km SSMF. If the phase encoding technique is applied to the optical pulse, the transmission distance may further increase. In addition, the optical pulse source can be implemented by a ring modulator (instead of an MZM) integrated into a silicon photonics chip [35]. Since the ring modulator is attractive in terms of footprint, power consumption, and bandwidth [35], it is necessary to demonstrate the feasibility of utilizing a ring modulator-integrated transmitter.

The high insertion loss of the transmitter is one of the disadvantages of the system. Although EDFAs were used in the previous experiments [21,30,34], we believe that a cost-effective SOA is required to make this system more attractive, such that it is cheap and integrable with other components, including a PD [4]. Then, the SOA would be located on the receiver side since the input power should be reduced to reduce the BER degradation (especially for the PAM-*N* signal) due to the nonlinear transfer characteristics arising from gain saturation [36,37]. The large noise figure of an SOA (compared to an EDFA) should also be considered [37]. Therefore, it is necessary to demonstrate a transmission experiment using a SiP transmitter and a receiver integrating an SOA and a PD.

As described in Section 3, robustness to CD is crucial to a high-speed transmission system. In the previous experiments, the transmission distances of the 150 Gb/s and 200 Gb/s OTDM signals were limited to <2 km (in the C-band region) even when using the electrical equalizer because the electrical equalizer cannot directly compensate for the CD in an IM/DD system [1]. Thus, it is worth developing ways to compensate for the CD in an optical domain to further increase the transmission distance (even if the O-band region is utilized). However, the use of the DCF is not considered an attractive solution for short-reach systems because of its large footprint [1]. Instead, it is necessary to demonstrate transmission experiments using other optical CD compensation methods, such as the use of a delay-line-based optical equalizer [38] or the fiber Bragg grating [39].

## 5. Conclusions

With a multilevel PAM modulation format, it is anticipated that 800 gigabit and 1.6 terabit Ethernet will be deployed soon. Considering the SINAD, the modulation level of PAM would not exceed eight. To further increase the line speed despite the limitation, a two-channel OTDM system was proposed, which is reviewed in this paper. In the system, the two-channel multiplexed signals are generated using a sinusoidal pulse and detected using a single PD. Compared to the four-channel OTDM system, the optical pulse can be simply generated using a modulator. We believe that this system is commercially deployable by integrating the transmitter into a single chip. Research has demonstrated that a transmitter consisting of three 17.2 GHz MZMs could generate the 150 Gb/s OTDM-PAM-8 signal when the FEC limit was set to $3.8\times 10^{-3}$. The generated signal was transmitted over a 1.9 km SSMF in the C-band wavelength region. The BER performance could be further improved by using an FFE-based demultiplexer (which is the first layer of a multilayer equalizer) and a nonlinear MIMO equalizer (which is the second layer equalizer). In addition, the use of a phase-alternating pulse allows for increasing the transmission distance. By replacing a constant-phase pulse with a phase-alternating pulse, the transmission distance of the 200 Gb/s OTDM-PAM-4 signal (the FEC limit was set to a BER of $3.8\times 10^{-3}$) increased from 1.0 km to 1.9 km (in the C-band region). Thus, we believe that the two-channel OTDM system can be utilized to generate a 400 Gb/s/λ (when using modulators with >30 GHz bandwidth) with sufficient transmission distance.

## Figures and Tables

**Figure 1 sensors-23-05908-f001:**
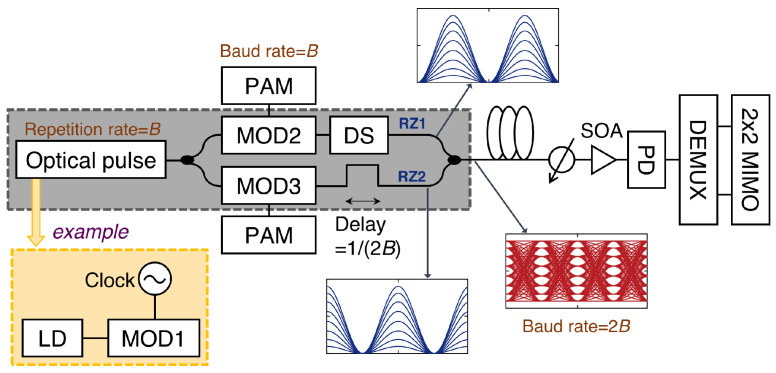
Schematic diagram of two-channel OTDM system (MOD: modulator, DEMUX: demultiplexer).

**Figure 2 sensors-23-05908-f002:**
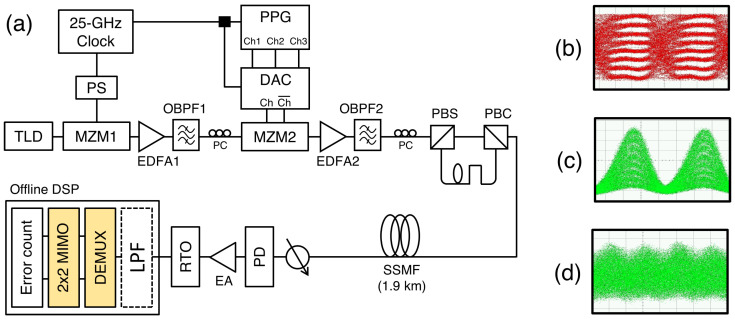
(**a**) Experimental setup for the transmission of a 150 Gb/s OTDM-PAM-8 signal. Eye diagrams of (**b**) a 75 Gb/s electrical PAM-8 signal, (**c**) a 75 Gb/s optical RZ-PAM-8 signal, and (**d**) a 150 Gb/s OTDM-PAM-8 signal. (TLD: tunable laser diode; OBPF: optical band-pass filter; PC: polarization controller; PS: phase shifter; EA: electrical amplifier; LPF: low-pass filter).

**Figure 3 sensors-23-05908-f003:**
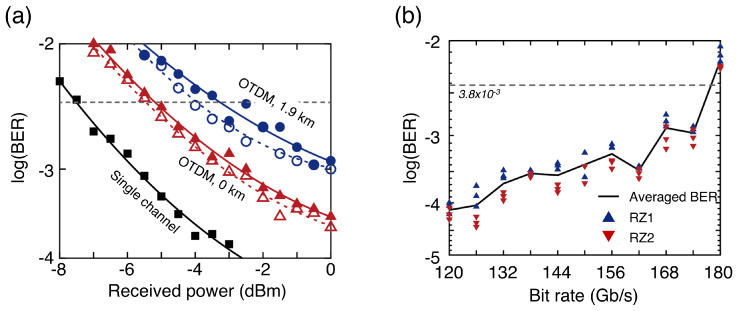
(**a**) BER curves of a 75 Gb/s PAM-8 signal generated by a single MZM and a 150 Gb/s OTDM-PAM-8 signal. (**b**) BERs of OTDM-PAM-8 signals as a function of bit rate.

**Figure 4 sensors-23-05908-f004:**
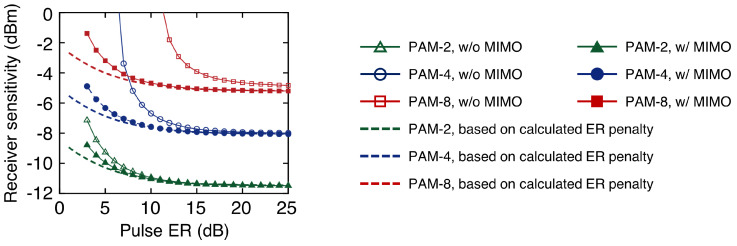
Receiver sensitivities (@BER = $3.8\times 10^{-3}$) of 150 Gb/s OTDM-PAM-*N* signals depending on the pulse ER.

**Figure 5 sensors-23-05908-f005:**
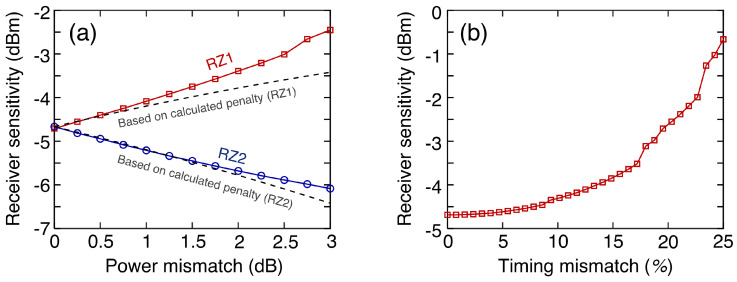
Receiver sensitivities of 150 Gb/s OTDM-PAM-8 signals depending on the (**a**) power difference and (**b**) timing difference between two RZ signals.

**Figure 6 sensors-23-05908-f006:**
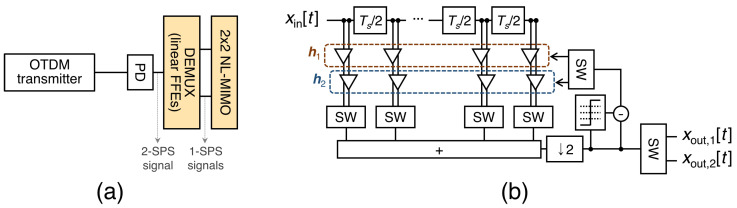
(**a**) Schematic diagram of a two-channel OTDM system using a multilayer equalizer. (**b**) Structure of the demultiplexer as the first-layer equalizer. (SPS: samples per symbol; NL: nonlinear; SW: switch).

**Figure 7 sensors-23-05908-f007:**
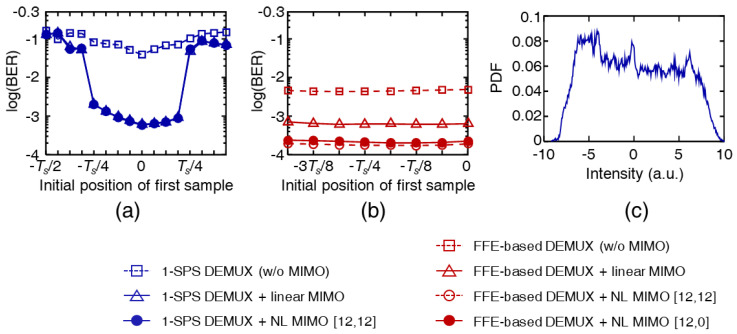
BERs depending on the sampling position when using (**a**) a symbol-spaced demultiplexer with a MIMO equalizer and (**b**) a multilayer equalizer. The received optical power was −1 dBm under the back-to-back condition. (**c**) Intensity histogram measured at the optimal timing phase when using a symbol-spaced demultiplexer.

**Figure 8 sensors-23-05908-f008:**
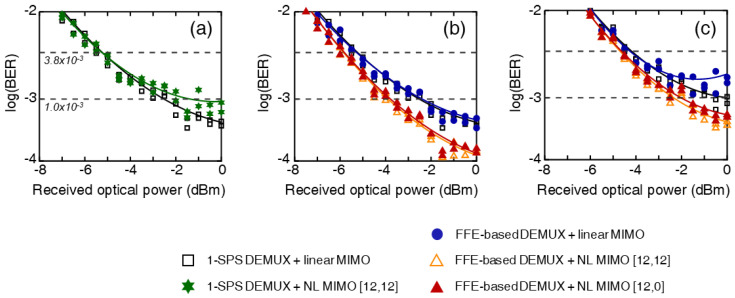
BER curves of the 150 Gb/s PAM-8 signals under the back-to-back condition when (**a**) a symbol-spaced demultiplexer with a MIMO equalizer and (**b**) a multilayer equalizer. (**c**) BER curves after 1.9 km transmission.

**Figure 9 sensors-23-05908-f009:**
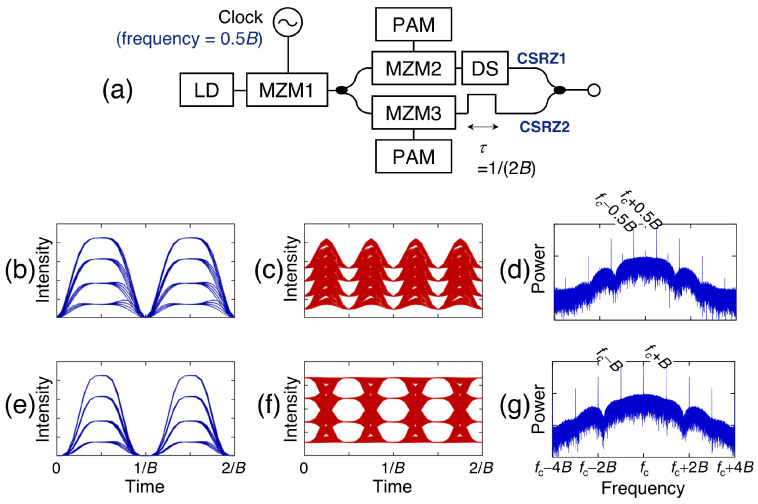
(**a**) OTDM transmitter consisting of a phase-alternating pulse generator and an MZM. Eye diagrams of the (**b**) CSRZ-PAM-4 signal; (**c**) OTDM-PAM-4 signal based on the CSRZ pulse; (**e**) RZ-PAM-4 signal; and (**f**) OTDM-PAM4 signal based on the RZ pulse. Optical spectra of the OTDM-PAM-4 signal generated using a (**d**) CSRZ pulse and (**g**) RZ pulse.

**Figure 10 sensors-23-05908-f010:**
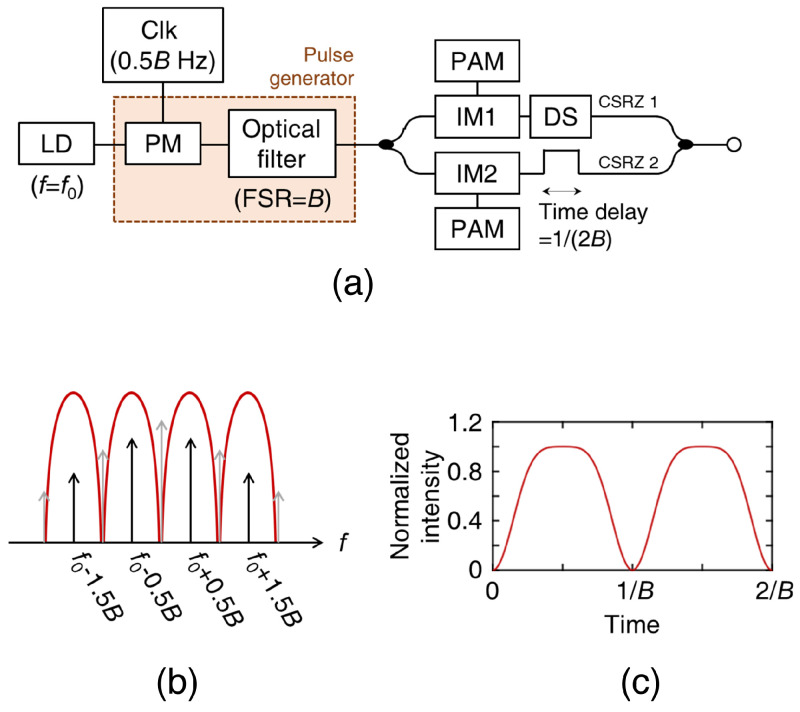
(**a**) OTDM transmitter with a phase-alternating pulse generated by a PM and a periodic optical filter. (**b**) Optical spectrum and (**c**) timing diagram of the CSRZ pulse. The ΔV of the PM is set to $0.5V_{\pi }$.

**Figure 11 sensors-23-05908-f011:**
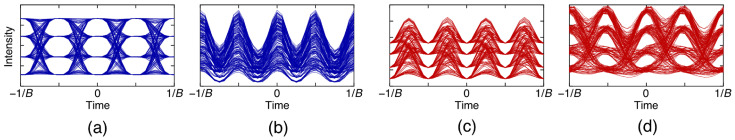
Eye diagrams of the 200 Gb/s OTDM-PAM-4 signals generated using a constant-phase pulse (**a**) before and (**b**) after transmission over 1 km SSMF and of the signal generated using a phase-alternating pulse (**c**) before and (**d**) after transmission over 1 km SSMF.

**Figure 12 sensors-23-05908-f012:**
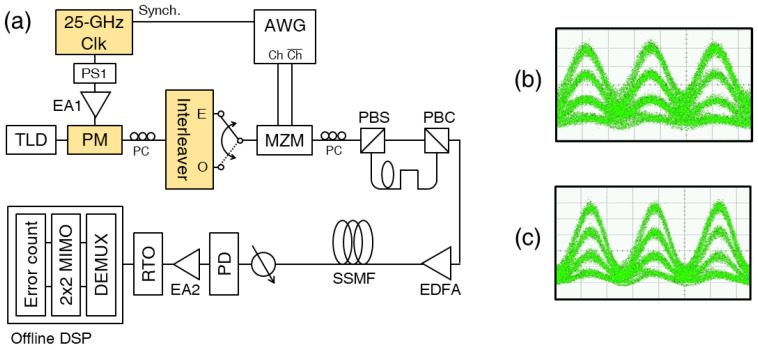
(**a**) Experimental setup for the transmission of 200 Gb/s OTDM-PAM-4 signal. Eye diagrams of the 100 Gb/s (**b**) CSRZ- and (**c**) RZ-PAM-4 signals.

**Figure 13 sensors-23-05908-f013:**
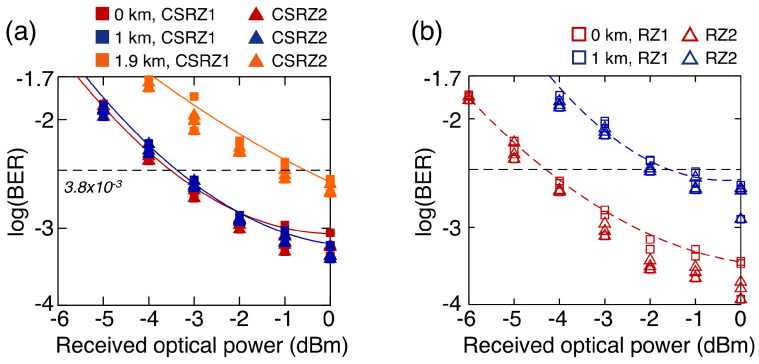
BER curves of the 200 Gb/s OTDM-PAM-4 signal generated using the (**a**) CSRZ pulse and (**b**) RZ pulse.

**Figure 14 sensors-23-05908-f014:**
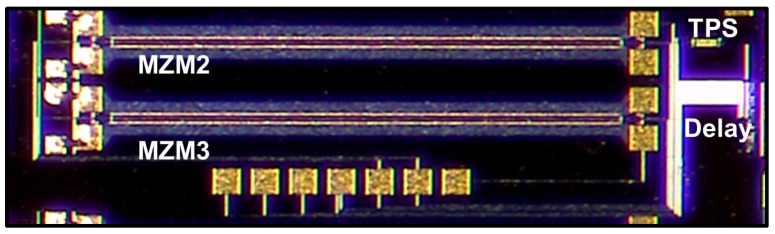
Photograph of an OTDM transmitter fabricated on a SiP chip.

## Data Availability

The data presented in this study are available on request from the corresponding author.

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
