# Peer review of "Two-Channel OTDM System for Data-Center Interconnects: A Review"

_sensors, 2023, doi:10.3390/s23135908_

Round 1
Reviewer 1 Report
The manuscript does a detailed survey of the two-channel OTDM system. It's logical and well-analyzed, but a few details need to be checked over, like line 28, where "DCIs" first appears, which connects in its full name "Data-Center Interconnects." In addition, authors should avoid quoting their papers extensively.
Reviewer 2 Report
This paper reviews the current stage of research on a two-channel optical time-division multiplexed system and discusses the possible research direction. However, I think that major revisions are required prior to publish this paper. To this end, the reviewer has the following comments:
1. The experimental conditions and system parameters are not given, and this is necessary for evaluating the effect of the method.
2. There is no equivalence comparison between experiment and practical application.
3. The authors have not provided the references for most of the equations.
4. The quality of figures needs to be improved, and the size of font is very small.
5. There are no horizontal or vertical coordinates in the figures, such as Figure 9, Figure 11 .
English needs to be improved
Reviewer 3 Report
After a careful review of the review paper entitled: "Two-Channel OTDM System for Data-Center Interconnects: A Review", I can metion that the manuscript is complete interesting from the point of view of current alternatives to improve the performances where the two-channel optical time-division multiplexed system can take place. Therefore, I recommend this manuscript for publication in Sensors after the following comments:
1. Is there any possibility that this kind of system can be integrated to enhance the actual progress of wireless optical communication process? If yes, how can be upgraded?
2. What kind of materials are involved into the two-channel OTDM systems?
Minor english errors should be addressed along the text.
Round 2
Reviewer 2 Report
English should be improved.
English should be improved.